# Fine-Tuning Pre-Trained Language Models Effectively by Optimizing Subnetworks Adaptively

**Haojie Zhang[1], Ge Li[1]∗, Jia Li[1], Zhongjin Zhang[1], Yuqi Zhu[1], Zhi Jin[1]**

[1]Key Laboratory of High Confidence Software Technologies (Peking University),
Ministry of Education; Institute of Software, EECS, Peking University, Beijing, China
`zhanghaojie@stu.pku.edu.cn, lige@pku.edu.cn`
`lijia@stu.pku.edu.cn, zjz123@stu.pku.edu.cn, zhuyuqi97@gmail.com, zhijin@pku.edu.cn`

## Abstract

Large-scale pre-trained language models have achieved impressive results on a wide range of downstream tasks recently. However, fine-tuning an extremely large-scale pre-trained language model on limited target datasets is often plagued by overfitting and representation degradation. In this paper, we propose a Dynamic Parameter Selection (DPS) algorithm for the large-scale pre-trained models during fine-tuning, which adaptively selects a more promising subnetwork to perform staging updates based on gradients of back-propagation. Experiments on the GLUE benchmark show that DPS outperforms previous fine-tuning methods in terms of overall performance and stability, and consistently achieves better results with variable pre-trained language models. In addition, DPS brings a large magnitude of improvement in out-of-domain transferring experiments and low-resource scenarios, which shows that it can maintain stable general contextual features and reduce the representation collapse. We release our code at `https://github.com/ZhangHaojie077/DPS`.

## 1 Introduction

The paradigm of pre-training on a large-scale unlabeled corpus and fine-tuning on task-specific datasets has led to a remarkable breakthrough in the field of Natural Language Processing (NLP) [Devlin et al., 2018, Raffel et al., 2019, Clark et al., 2020, He et al., 2021]. While the great success has been achieved on various downstream tasks, previous research has pointed out that such approaches are still facing two limitations: ❶ Fine-tuning instability. With the same hyper-parameters, different restarts will lead to huge differences in the performance of pre-trained language models, especially on small datasets [Phang et al., 2018, Dodge et al., 2020, Lee et al., 2020, Zhang et al., 2021, Mosbach et al., 2021]. ❷ Poor generalizability. Limited by the size of the downstream datasets, aggressive fine-tuning can lead to catastrophic forgetting and overfitting, which causes the model to perform poorly when transferring to out-of-domain data or other related tasks [Mahabadi et al., 2021, Aghajanyan et al., 2021, Xu et al., 2021]. Therefore, it remains a challenging topic to adapt large-scale pre-trained models to various scenarios while maintaining their stability and generalizability to the fullest.

Optimizing a subnetwork of pre-trained models has been proved to be an effective approach to improve stability or mitigate overfitting, such as Mixout [Lee et al., 2020] and CHILD-TUNING$_D$ [Xu et al., 2021]. Mixout utilizes pre-trained weights to randomly replace some parameters with a probability of $p$ and updates only the subnetwork composed of the parameters without replacement. CHILD-TUNING$_D$ computes the importance of parameters based on all training samples' gradients before training and chooses an unchanged subnetwork composed of important parameters for updating.

---

∗Corresponding author.

36th Conference on Neural Information Processing Systems (NeurIPS 2022).

Despite their great performance, we argue that (i) choosing subnetworks randomly (Mixout) ignores the importance of parameters; (ii) choosing an unchanged subnetwork and ignoring the update of some parameters (CHILD-TUNING$_D$) may affect the model's perception of downstream tasks. Moreover, accumulating all training samples' gradients calculated by back-propagation outside training would induce a heavy additional computational overhead.

To overcome the above two restrictions of existing subnetwork optimizations, we propose a Dynamic Parameter Selection (DPS) algorithm to fine-tune pre-trained models adaptively. Overall, DPS consists of a **cyclic two-stage** updating strategy: Stage I accumulates in-batch gradients of updated parameters during fine-tuning; Stage II utilizes accumulated values of Stage I to derive a stage-specific subnetwork composed of important parameters to update while keeping non-subnetwork constant. Note that both two stages consist of several iterations and the whole fine-tuning process will repeat several two-stage updating processes until the model convergence. Compared to previous subnetwork optimizations, we summarize the advantages of DPS as follows: (i) DPS can adaptively choose out more promising subnetworks based on the changing parameter importance while paying attention to more parameters possibilities, which enables better optimization of the model. (ii) DPS can effectively combine two processes: fine-tuning and deriving subnetworks, which greatly reduces the additional heavy computational costs introduced by separating above two processes like CHILD-TUNING$_D$.

In details, we propose two variants, DPS Dense and DPS Mix, to optimize the fine-tuning process. DPS Dense aims at optimizing the full network update process. In Stage I, DPS Dense updates the whole network while accumulating gradients' values. In Stage II, DPS Dense updates the stage-specific subnetwork composed of important parameters measured by ranking the accumulated values of Stage I. In contrast, DPS Mix is designed to optimize the stochastic parameter update process. In Stage I, DPS Mix randomly replaces some outgoing parameters of the neural network with pre-trained parameters and accumulates in-batch gradients of the remaining updated parameters. In Stage II, DPS Mix applies a frequency-penalized selection strategy, which utilizes updated gradients' average value and an exponential penalty factor negatively correlated with update frequency to choose out the stage-related important parameters for updating. It adaptively chooses a more promising subnetwork in stages while avoiding deviations from the pre-trained weights due to aggressive optimization.

We re-evaluate several state-of-the-art fine-tuning regularization approaches [Lee et al., 2020, Aghajanyan et al., 2021, Zhang et al., 2021, Xu et al., 2021, Wu et al., 2021] on GLUE Benchmark and find that DPS outperforms prior approaches by $0.44 \sim 1.33$ points gains and $0.29 \sim 0.80$ std (standard deviation) drops. Moreover, DPS brings an improvement in out-of-domain transferring experiments/low-resource scenarios with up to 1.86/3.27 average score, indicating that DPS can obtain excellent generalization ability. In extensive experiments, we further explore DPS in the following four settings: (i) subnetwork size; (ii) various pre-trained models; (iii) a sufficient number of training examples; (iv) time usage. Based on the above extensive experiments, we observe that (i) DPS can better capture task-relevant parameters across a variety of subnetwork sizes than Mixout and CHILD-TUNING$_D$; (ii) DPS can consistently bring a stable improvement in various pre-trained models; (iii) our proposed DPS is helpful to model performance in fine-tuning with a sufficient number of training examples; (iv) DPS is a more time-efficient approach than several fine-tuning methods.

## 2 Approach

### 2.1 Background and Related Work

We first start to introduce the paradigm of fine-tuning the subnetwork by giving formulations of forward and back propagation during standard fine-tuning, Mixout, and CHILD-TUNING$_D$. We will describe all formulations for Stochastic Gradient Descent (SGD) iterations. We assume $W^{(t)} = (W_1^{(t)}, \cdots, W_n^{(t)})$, where $W_i^{(t)}$ represents parameters of the $i$-th neuron at the $t$-th iteration. Standard fine-tuning formulates model parameters as follows:

$$W^{(t+1)} = W^{(t)} - \eta \frac{\partial \mathcal{L}\left(W^{(t)}\right)}{\partial W^{(t)}}, \tag{1}$$

where $\mathcal{L}$ represents in-batch loss; $\eta$ represents learning rate. Differ from fine-tuning whole network, Mixout first derives a mask matrix $M^{(t)} = (M_1^{(t)}, \cdots, M_n^{(t)})$, where $M_i^{(t)}$ is same-sized as $W_i^{(t)}$ and

$M_i^{(t)}$'s are mutually independent $Bernoulli(1-p)$ for all $i$ and $t$. Then Mixout replaces parameters corresponding to zero values in $M^{(t)}$ with pre-trained parameters and only update the subnetwork composed of remaining parameters. Its formulation is as follows:

$$W^{(t+1)} = W^{(t)} - \eta \frac{\partial \mathcal{L}\left(\left(M^{(t)}W^{(t)} + \left(I - M^{(t)}\right)W^{(0)} - pW^{(0)}\right)(1-p)^{-1}\right)}{\partial W^{(t)}}, \qquad (2)$$

where $I$ is full-one matrix and is same-sized as $M^{(t)}$; $W^{(0)}$ represents the parameter matrix of the pre-trained model; the larger $p$ is, the smaller the subnetwork.

Besides selecting a subnetwork randomly, CHILD-TUNING$_D$ has explored fine-tuning subnetwork composed of measured important parameters. Considering Fisher Information [Fisher, 1922] serves as a good way of measuring the amount of information that a random variable carries about a parameter of the distribution [Tu et al., 2016] and has been widely used in modern machine learning and deep learning [Achille et al., 2019, Kirkpatrick et al., 2017, Amari, 1996, Pascanu and Bengio, 2013, Singh and Alistarh, 2020, Crowley et al., 2018, Martens, 2014, Theis et al., 2018], CHILD-TUNING$_D$ utilizes Fisher Information to compute the importance of parameters based on all training samples before training, then derives an unchanged subnetwork composed of important parameters to update. Its formulation is as follows:

$$M_{CT_D} = \Sigma(\frac{\partial \mathcal{L}\left(W^{(0)}\right)}{\partial W^{(0)}})^2 > \text{sort}\left(\Sigma(\frac{\partial \mathcal{L}\left(W^{(0)}\right)}{\partial W^{(0)}})^2\right)_p, \qquad (3)$$

$$W^{(t+1)} = W^{(t)} - \eta \frac{\partial \mathcal{L}\left(W^{(t)}\right)}{\partial W^{(t)}} M_{CT_D}, \qquad (4)$$

where $sort(A)_p$ represents the value of $p$ percentile in matrix A after sorting in ascending order; $M_{CT_D}$ is an unchanged mask matrix, which determines the subnetwork used for updates throughout the fine-tuning process.

Since their introduction that aggressive fine-tuning will lead to instability and overfitting, especially on small datasets[Devlin et al., 2018, Phang et al., 2018], various fine-tuning methods have been proposed to solve the above problems, except for subnetwork optimizations. Some of them utilize output representations or inject noise into input to regularize models [Mahabadi et al., 2021, Aghajanyan et al., 2021, Wu et al., 2021], while others adjust the weights of top layers of Transformer or BERTAdam optimizer to make the pre-trained models more task-relevant. [Zhang et al., 2021, Mosbach et al., 2021].

---

**Algorithm 1** Training Algorithm for DPS

---

**Require: ms**: total training steps; **ur**: update ratio; **us**: update steps; **p**: the proportion of parameters not updated to the overall parameters in Stage II; **k**: current cycle times; **n**: total cycle times

1: **Initialize** timestep $t \leftarrow 1$, **us** $\leftarrow$ **ms** $*$ **ur**, **n** $\leftarrow \lfloor$**ms/2us**$\rfloor$
2: **for k** $= 0$ to **n do**
3:     // $k$-th cycle
4:     **while** $t \leq (\mathbf{k}+1) * 2\mathbf{us}$ **do**
5:         // Stage I
6:         **if** $\lceil t/\mathbf{us}\rceil \,\%2 = 1$ **then**
7:             Accumulate in-batch gradients of the updated parameters during fine-tuning.
8:         //Stage II
9:         **if** $\lceil t/\mathbf{us}\rceil \,\% \,2 = 0$ and $(t-1)\,\%\,\mathbf{us} = 0$ **then**
10:             Derive a stage-specific mask matrix by selecting the top $(1 - \mathbf{p})$ parameters by a **specified selection strategy**.
11:             Clear the accumulated values of Stage I.
12:             Update the selected subnetwork by stage-specific mask matrix.
13:         **else**
14:             Update the selected subnetwork by stage-specific mask matrix.
15:         $t \leftarrow t + 1$
16:     **end while**
17: **end for**

---

## 2.2 Overview of DPS

DPS consists of a cyclic two-stage updating strategy: Stage I accumulates in-batch gradients of updated parameters during fine-tuning; Stage II utilizes accumulated values of Stage I to derive a stage-specific subnetwork composed of important parameters to update while keeping non-subnetwork constant. We provide a pseudo-code of DPS in Algorithm 1. We argue that fine-tuning a dynamic promising subnetwork while paying attention to more parameters possibilities can align all parameters more relevant for a specific downstream task, which in turn better unlocks the potential of the model. To select a promising subnetwork adaptively, we propose two enhanced fine-tuning strategies: DPS Dense (in Sec. 2.3) and DPS Mix (in Sec. 2.4), which respectively optimize the fine-tuning processes of full network update and stochastic network update.

## 2.3 DPS Dense

We firstly explore the DPS in full network update called DPS Dense. Before fine-tuning, DPS Dense initializes a Gradient Accumulation Matrix ($GAM$), which is the same-sized as $W^{(0)}$. In Stage I, DPS Dense fine-tunes the whole network as Equation 1 while accumulating the gradients' square of all parameters at every iteration with the help of $GAM$; In Stage II, DPS Dense first ranks $GAM$ and derives a stage-specific mask matrix $M_{DPS}$ filtered by selecting top $(1-p)$ highest Fisher Information of $GAM$, then utilizes $M_{DPS}$ to update the stage-specific subnetwork composed of important parameters. we denote the formulation of Stage II as follows:

$$M_{DPS} = GAM > \text{sort}(GAM)_p, \tag{5}$$

$$W^{(t+1)} = W^{(t)} - \eta \frac{\partial \mathcal{L}\left(W^{(t)}\right)}{\partial W^{(t)}} M_{DPS}. \tag{6}$$

The specific update process can be seen in Figure 1.

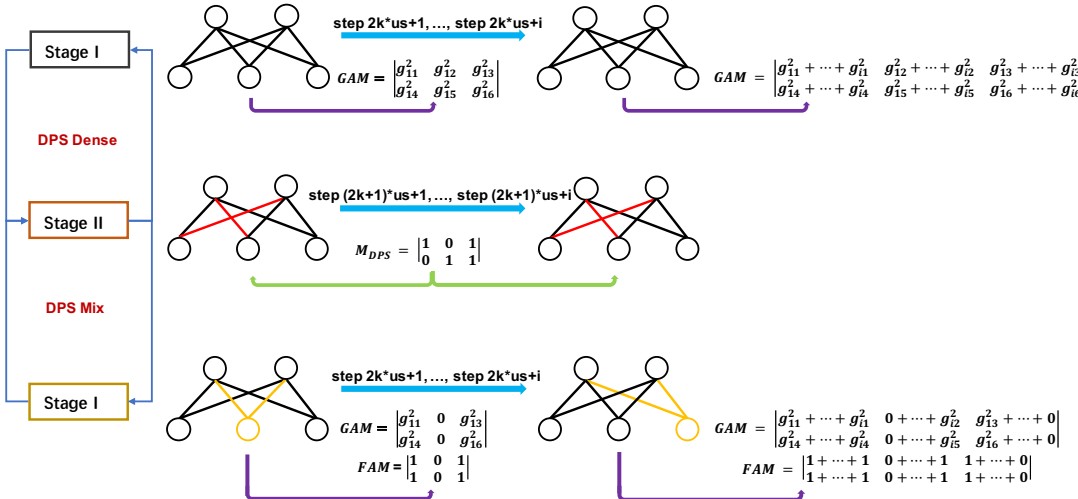

Figure 1: In this figure, we show the k-th two-stage updating process of DPS Dense and DPS Mix. $g_{ij}$ represents the the gradient value of parameter $j$ at time step $2k*us+i$ ($i <= us$); the parameters on the red edges are same-valued as corresponding parameters of $W^{((2k+1)us+1)}$, the parameters on the orange edges are same-valued as corresponding parameters of $W^{(0)}$. Both red and orange edges will not be updated during back propagation; $GAM$, $FAM$ and $M_{DPS}$ will be re-initialized at the beginning of every cycle.

## 2.4 DPS Mix

Considering stochastic parameter update is common to deploy in fine-tuning pre-trained models, we also propose an alternative to optimize this fine-tuning process called DPS Mix. Before fine-tuning, DPS Mix needs to initialize another Frequency Accumulation Matrix ($FAM$) besides $GAM$, which

is used to store the frequency of every parameter being updated in Stage I. In Stage I, DPS Mix derives a mask matrix $M^{(t)}$ like Mixout, and then fine-tunes the subnetwork determined by $M^{(t)}$ while replacing non-subnetwork as pre-trained weights as Equation 2. In Stage II, DPS Mix derives $M_{DPS}$ according to a frequency-penalized selection strategy as follows:

$$M_{DPS} = \begin{cases} \left(\frac{GAM}{FAM}\right) > \text{sort}\left(\frac{GAM}{FAM}\right)_p, \text{ if } us < 50 \\ \left(\frac{GAM}{FAM}e^{-\left(\frac{FAM}{us}\right)}\right) > \text{sort}\left(\frac{GAM}{FAM}e^{-\left(\frac{FAM}{us}\right)}\right)_p, \text{ else} \end{cases}. \qquad (7)$$

As Equation 7 illustrates, DPS Mix follows the ratio of $GAM$ and $FAM$ as the basis for deriving the $M_{DPS}$. During the random selection of Stage I, considering that each parameter may be updated at an inconsistent frequency and parameters with more frequent updates may have smaller potential, DPS Mix also multiplies an exponential penalty factor negatively correlated with update frequency in addition to the ranking selection, to give more chances to parameters with fewer updates. Since this penalty factor may introduce some fluctuations, we grid search the boundary and decide to exclude this factor when $us$ is less than 50. After obtaining the $M_{DPS}$, DPS Mix formulates model parameters as follows:

$$W^{(t+1)} = W^{(t)} - \eta \frac{\partial \mathcal{L}\left(\left(M_{DPS}W^{(t)} + (I - M_{DPS})W' - pW'\right)(1-p)^{-1}\right)}{\partial W^{(t)}}, \qquad (8)$$

where $W'$ are same-valued as $W^{((2k+1)us+1)}$; Note that the subnetwork size of Stage I is the same-sized as Stage II's in DPS Mix. The specific update can be seen in Figure 1. In terms of implementation details, we apply DPS Dense and DPS Mix to specific layers.

# 3 Experiments

## 3.1 Datasets

**GLUE benchmark.** Following previous studies [Lee et al., 2020, Dodge et al., 2020, Zhang et al., 2021], we conduct a series of extensive experiments on eight datasets from the GLUE benchmark [Wang et al., 2019]. The datasets cover four tasks: natural language inference (RTE, QNLI, MNLI), paraphrase detection (MRPC, QQP), linguistic acceptability (CoLA), and sentiment classification (SST-2). Appendix B provides dataset statistics and a brief description of each dataset. Since the test set can only be submitted online for a very limited times, we follow several previous studies [Phang et al., 2018, Lee et al., 2020, Dodge et al., 2020, Aghajanyan et al., 2021, Zhang et al., 2021, Xu et al., 2021] that fine-tune on the training sets and report the results on the development sets.

**NLI Datasets.** We evaluate and probe the generalization ability of DPS on several Natural Language Inference (NLI) tasks, including SNLI [Bowman et al., 2015], MNLI [Williams et al., 2018], MNLI-M [Williams et al., 2018], RTE [Bentivogli et al., 2009], SICK [Marelli et al., 2014] and SciTail [Khot et al., 2018]. All tasks are reported by Accuracy. Since the target datasets have different label spaces, during the evaluation, we map predictions to each target dataset's space (in Appendix E).

## 3.2 Experimental Setup

We use the pre-trained models and codes provided by HuggingFace* Wolf et al. [2020]. Appendix C provides specific hyper-parameter details, and unless noted otherwise, we follow the default hyper-parameter setup of HuggingFace.

## 3.3 Comparing Prior Methods for Few-Sample BERT Fine-tuning

### 3.3.1 Baseline

**Mixout** [Lee et al., 2020] is a stochastic parameter update technique based on the Bernoulli distribution for preventing catastrophic forgetting during fine-tuning. It randomly replaces the model parameters with pre-trained parameters with a probability of $p$. **R3F** [Aghajanyan et al., 2021] is motivated by trust region theory and injects parametric noise to the embedding of the pre-trained

---

*https://github.com/huggingface/transformers

representations. **R-Dropout** [Wu et al., 2021] minimizes the bidirectional KL-divergence between the output distributions of two sub models sampled by Dropout. **CHILD-TUNING**$_D$ [Xu et al., 2021] utilizes the downstream task data to detect the most task-related parameters as the child network and freezes the parameters in the non-child network to their pre-trained weights during fine-tuning. **Re-init** [Zhang et al., 2021] re-initializes the pooler layers and the top $L$ BERT Transformer layers using the original BERT initialization, $\mathcal{N}(0,0.02^2)$.

### 3.3.2 Results

In this section, we compare DPS with a variety of fine-tuning regularization approaches on BERT$_{LARGE}$, following [Lee et al., 2020, Zhang et al., 2021, Xu et al., 2021]. The hyper-parameter search spaces of different fine-tuning regularization methods are supplemented in Appendix D. Table 1 summarizes the experimental results. Firstly, we can notice that Mixout and CHILD-TUNING$_D$ achieve better results than vanilla fine-tuning, and perform nearly the best baseline Re-init, which proves that fine-tuning a subnetwork while utilizing pre-trained weights is a competitive approach to regularize the model. Secondly, our proposed straightforward DPS Dense outperforms all baselines on average, which demonstrates that fine-tuning subnetworks adaptively is more effective. Since DPS accumulates in-batch gradients to derive important sub-networks during fine-tuning, we also consider DPS to be a more time-efficient approach than CHILD-TUNING$_D$, which utilizes Fisher Information to derive important sub-network based on all training samples outside fine-tuning. Thirdly, it is easy to find that DPS Mix achieves better results while further improving stability, especially on CoLA with great std drop. This reveals that DPS can be effectively compatible with various fine-tuning strategies, enabling better optimization of the model.

| Methods | CoLA | | MRPC | | RTE | | STS-B | | Avg. | |
|---|---|---|---|---|---|---|---|---|---|---|
| | mean | std | mean | std | mean | std | mean | std | mean | std |
| vanilla fine-tuning | 64.11 | 1.33 | 90.80 | 1.77 | 70.69 | 2.83 | 89.92 | 0.61 | 78.88 | 1.64 |
| Mixout | 64.42 | 1.51 | 91.31 | 1.08 | 72.05 | 1.67 | 90.39 | 0.57 | 79.54 | 1.21 |
| R3F | 64.62 | 1.38 | 91.63 | 0.93 | 70.75 | 1.76 | 89.92 | 0.61 | 79.23 | 1.17 |
| R-Dropout | 64.14 | 1.58 | **91.87** | 0.78 | 70.24 | 2.83 | 90.25 | 0.49 | 79.13 | 1.42 |
| CHILD-TUNING$_D$ | 64.85 | 1.32 | 91.52 | 0.81 | 71.69 | 1.95 | 90.42 | 0.44 | 79.62 | 1.13 |
| Re-init | 64.24 | 2.03 | 91.61 | 0.80 | 72.44 | 1.74 | **90.71** | **0.14** | 79.77 | 1.18 |
| DPS Dense | 64.98 | 1.08 | 91.50 | 0.83 | 73.14 | 1.97 | 90.51 | 0.55 | 80.03 | 1.11 |
| DPS Mix | **65.11** | **0.63** | 91.78 | **0.74** | **73.46** | **1.46** | 90.47 | 0.55 | **80.21** | **0.84** |

Table 1: We report the mean and standard deviation results of 10 random seeds. Avg. represents the average score of the four tasks, the best results are bolded. The higher the mean value, the better the effect; the smaller the standard deviation, the more stable the performance. Since R3F is not applicable to the regression task (STS-B), we use the results of vanilla fine-tuning as the results of R3F on STS-B.

### 3.4 Results on Out-of-Domain Generalization

Previous research has found that models trained on annotated datasets create superficial shortcut features and do not generalize well to out-of-domain datasets [Belinkov et al., 2019]. Since DPS possesses strong and stable performance, we also expect DPS can help to discover deeper semantic features and achieve better generalization in transferring to out-of-domain datasets or other related tasks. In detail, to evaluate out-of-domain generalization, we fine-tune BERT$_{LARGE}$ on medium-sized 8K subsampled MNLI and SNLI datasets respectively, and directly evaluate the fine-tuned models on several NLI datasets, including MNLI, MNLI-M, SICK, SciTail, SNLI, and RTE. The experimental results are shown in Table 2. On the models trained on MNLI, DPS Dense/DPS mix outperforms vanilla fine-tuning by 1.16/1.86 average score, and consistently outperforms CHILD-TUNING$_D$ by 0.32/1.05 average score. On the models trained on SNLI, although CHILD-TUNING$_D$ only brings a very slight improvement over vanilla fine-tuning, DPS Dense/DPS mix consistently delivers a large improvement of 1.23/0.63 average score. All the above statistics can fully verify that DPS can better explore deeper semantics and reduce superficial biases unique to source task than vanilla fine-tuning and CHILD-TUNING$_D$. We also conclude that making the model closer to real distribution for the source task may facilitate generalization ability better than freezing some parameters as pre-trained weights like CHILD-TUNING$_D$.

| Datasets | MNLI | | | | SNLI | | | |
|---|---|---|---|---|---|---|---|---|
| | vanilla | $CT_D$ | DPS Dense | DPS Mix | vanilla | $CT_D$ | DPS Dense | DPS Mix |
| MNLI | 77.51 | **78.01** | 77.59 | 77.75 | 66.43 | 67.26 | **67.32** | 66.81 |
| MNLI-M | 78.75 | 79.13 | 79.03 | **79.16** | 68.07 | 68.75 | **69.08** | 68.47 |
| SICK | 51.91 | 55.69 | 55.76 | **58.18** | 53.67 | 53.46 | **56.23** | 55.49 |
| SciTail | 79.70 | 79.86 | 79.35 | **80.36** | 78.22 | 77.17 | **78.42** | 77.80 |
| SNLI | 72.80 | 73.36 | 74.28 | **74.72** | **84.87** | 84.41 | 84.74 | 84.83 |
| RTE | 71.44 | 70.87 | 72.80 | **73.16** | 67.35 | 67.47 | **70.28** | 69.07 |
| Avg. | 72.01 | 72.82 | 73.14 | **73.87** | 69.78 | 69.79 | **71.01** | 70.41 |

Table 2: All models are trained on MNLI or SNLI and evaluated on the out-of-domain datasets. We report the average results of three random seeds. $CT_D$ represents CHILD-TUNING$_D$.

## 3.5 Results on Low-Resource Scenarios

To further explore the generalization ability of DPS on extremely small datasets, we downsample eight datasets in GLUE to 0.5K and 1K, and fine-tune them all on BERT$_{LARGE}$. As illustrated in Table 3, DPS outperforms vanilla fine-tuning by a large margin almost on all tasks in variable low-resource scenarios, which demonstrates that DPS, as a novel fine-tuning technique, can effectively mitigate the risk of overfitting, exploit model potential, and remove irrelevant features.

| Datasets | 0.5K | | | | | | 1K | | | | | |
|---|---|---|---|---|---|---|---|---|---|---|---|---|
| | vanilla | | DPS Dense | | DPS Mix | | vanilla | | DPS Dense | | DPS Mix | |
| | mean | std | mean | std | mean | std | mean | std | mean | std | mean | std |
| CoLA | 26.01 | 13.2 | 39.42 | 11.8 | **40.58** | 10.6 | 47.97 | 5.62 | 52.89 | 2.50 | **53.34** | 2.35 |
| MRPC | 82.96 | 0.84 | 83.29 | 0.78 | **83.33** | 0.56 | 85.34 | 1.30 | **85.90** | 0.86 | 85.40 | 1.20 |
| RTE | 58.30 | 4.90 | 58.34 | 2.90 | **59.13** | 3.49 | 62.60 | 3.46 | 64.44 | 1.76 | **64.77** | 1.66 |
| STS-B | 81.77 | 2.69 | 83.38 | 1.55 | **83.60** | 1.18 | 85.86 | 1.34 | **87.15** | 1.77 | 87.02 | 1.04 |
| SST-2 | 86.51 | 6.48 | **89.38** | 0.52 | 89.07 | 0.38 | 90.28 | 0.55 | **90.92** | 0.64 | 90.84 | 0.58 |
| QNLI | 76.82 | 2.09 | **77.09** | 1.75 | 76.73 | 1.92 | 81.61 | 1.32 | 82.58 | 1.19 | **82.66** | 1.26 |
| QQP | 71.68 | 1.90 | **74.43** | 1.35 | 74.20 | 1.42 | 76.96 | 1.50 | **77.81** | 0.57 | 77.71 | 0.31 |
| MNLI | 42.81 | 4.49 | **46.61** | 2.70 | 46.28 | 2.64 | 54.93 | 5.94 | 58.33 | 3.56 | **58.53** | 3.27 |
| AVG | 65.85 | 4.57 | 68.99 | 2.92 | **69.12** | 2.78 | 73.19 | 2.62 | 75.00 | 1.61 | **75.04** | 1.45 |

Table 3: Comparison between DPS and vanilla fine-tuning with varying low-resource scenarios (0.5K,1K). We report the results of 10 random seeds and the best result of each task is bolded.

# 4 Analysis and Study

## 4.1 Effect of Subnetwork Size

[Lee et al., 2020] have explored the hyper-parameter $p$ and verified that Mixout shows great performance at many different probabilities $p$ (Note that the larger $p$, the smaller the subnetwork). We also wonder what will happen when DPS probabilities vary, so we plot the validation performance distribution of CoLA and RTE of 10 random seeds when fine-tuning BERT$_{LARGE}$ across different probabilities $p \in \{0.1, 0.2, 0.3, 0.4, 0.5\}$ in Figure 2. Firstly, it is easy to find the overall performance of DPS Dense/DPS Mix is better than CHILD-TUNING$_D$/Mixout across all probabilities, which demonstrates fine-tuning subnetworks of the pre-trained model adaptively can better optimize the model than fine-tuning an unchanged subnetwork or fine-tuning random subnetworks. Secondly, compared to vanilla fine-tuning, there is a visible performance drop in CHILD-TUNING$_D$ and Mixout with some probabilities. However, DPS Dense/DPS Mix consistently outperforms vanilla fine-tuning across all probabilities. These fully indicate that CHILD-TUNING$_D$ and Mixout may fail to capture important parameters in some sizes of the subnetwork, resulting in failed runs (we refer to the average scores lower than vanilla as failed runs). However, Our proposed DPS can better capture more task-relevant parameters, which in turn results in a consistent and stable improvement in all sizes of the subnetwork. To further justify the effectiveness of DPS from another perspective, we conduct a case study in Appendix A.

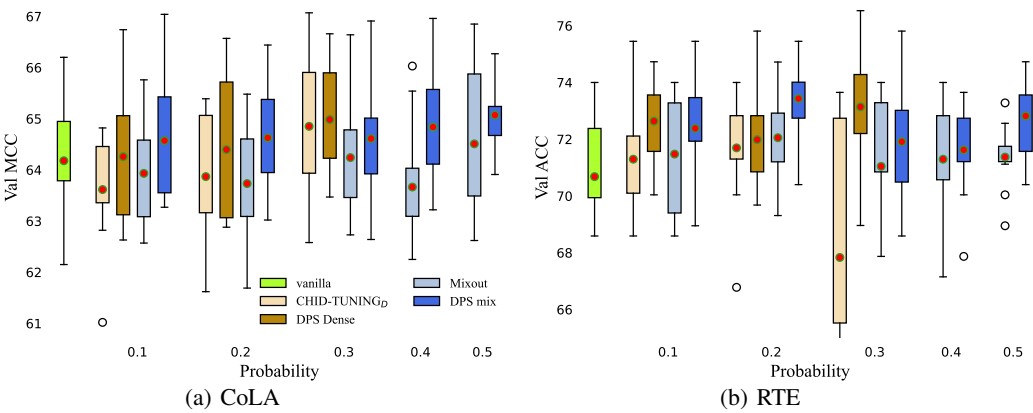

Figure 2: Validation performance distribution of 10 random seeds when fine-tuning BERT$_{LARGE}$ with vanilla fine-tuning, Mixout, CHILD-TUNING$_D$ and DPS. Solid circles represent mean results, hollow circles represent outliers.

## 4.2 Results on Different Pre-Trained Language Models

| Models | Methods | CoLA | | MRPC | | RTE | | STS-B | | Avg. | |
|---|---|---|---|---|---|---|---|---|---|---|---|
| | | mean | std | mean | std | mean | std | mean | std | mean | std |
| BERT | vanilla | 64.11 | 1.33 | 90.80 | 1.77 | 70.69 | 2.83 | 89.92 | 0.61 | 78.88 | 1.64 |
| | DPS Dense | 64.98 | 1.08 | 91.50 | 0.83 | 73.14 | 1.97 | 90.51 | **0.55** | 80.03 | 1.11 |
| | DPS Mix | **65.11** | **0.63** | **91.78** | **0.74** | **73.46** | **1.46** | 90.47 | **0.55** | **80.21** | **0.84** |
| RoBERTa | vanilla | 66.82 | **1.16** | 92.96 | 0.41 | 85.61 | **1.02** | 92.31 | **0.12** | 84.43 | **0.68** |
| | DPS Dense | 68.05 | 1.25 | 93.01 | 0.33 | **86.02** | 2.26 | **92.62** | 0.16 | 84.93 | 1.00 |
| | DPS Mix | **68.81** | 1.33 | **93.27** | **0.27** | 85.74 | 1.16 | 92.57 | 0.17 | **85.10** | 0.73 |
| BART | vanilla | 61.34 | 1.83 | 92.58 | 0.43 | 82.84 | 1.22 | 91.89 | 0.25 | 82.16 | 0.93 |
| | DPS Dense | **64.56** | 1.92 | 92.54 | 0.52 | 85.28 | 0.87 | 92.38 | 0.22 | 83.69 | 0.88 |
| | DPS Mix | 64.37 | **1.21** | **92.88** | **0.39** | **85.31** | **0.69** | **92.44** | **0.20** | **83.75** | **0.61** |
| ELECTRA | vanilla | 65.83 | 14.5 | 93.00 | 0.42 | 89.25 | 0.88 | 90.36 | 6.56 | 84.66 | 5.59 |
| | DPS Dense | 69.25 | 2.98 | **93.95** | 0.40 | 89.45 | 0.83 | **92.69** | 0.25 | 86.34 | 1.12 |
| | DPS Mix | **70.41** | **2.01** | 93.58 | **0.33** | **90.03** | **0.76** | 92.65 | 0.28 | **86.67** | **0.84** |
| DeBERTa | vanilla | 65.51 | 1.54 | 92.60 | 0.69 | 85.71 | 1.26 | 91.60 | 0.44 | 83.86 | 0.98 |
| | DPS Dense | **67.38** | 0.93 | **93.07** | **0.22** | 86.86 | 0.91 | **92.25** | 0.40 | **84.89** | 0.62 |
| | DPS Mix | 67.07 | **0.84** | 92.85 | 0.34 | **86.91** | **0.77** | 92.21 | **0.37** | 84.76 | **0.58** |

Table 4: We compare DPS and vanilla fine-tuning on five widely used PLMs and report the mean and standard deviation results of 10 random seeds. Avg. represents the average score of the four tasks, the best results are bolded.

In this subsection, we conduct experiments on the four tasks in GLUE on five different Pre-trained Language Models (PLMs): BERT$_{LARGE}$ [Devlin et al., 2018], RoBERTa$_{LARGE}$ [Liu et al., 2019], BART$_{LARGE}$ [Lewis et al., 2019], ELECTRA$_{LARGE}$ [Clark et al., 2020] and DeBERTa$_{LARGE}$ [He et al., 2021]. The experimental results are shown in Table 4. In addition to the most classical model BERT, DPS consistently achieves better results on both RoBERTa and ELECTRA trained on larger pre-trained corpus and better pre-training tasks. DPS also improves a large magnitude on models with various structures: DeBERTa (relative position encoding model) and BART (encoder-decoder model). These fully demonstrate that DPS can bring a stable improvement regardless of model structure and quality.

### 4.3 DPS with a Sufficient Number of Training Examples

In addition to few-sample datasets, we believe it is also worth exploring whether DPS can deliver a boost with larger training sets. Therefore, we fine-tune $\text{BERT}_{LARGE}$ on two datasets in GLUE with training samples over 50K, including SST-2 and QNLI. Since it has been stable to fine-tune $\text{BERT}_{LARGE}$ with a sufficient number of training examples Devlin et al. [2018], Phang et al. [2018], we focus on the expressiveness of the above two tasks. As the experimental results in the Table 5 noted, when the training samples are large enough, DPS not only doesn't damage performance but also brings certain improvements compared to other subnetwork optimizations. This indicates that our proposed dynamic parameter selection algorithm is suitable for a wider range of data scenarios, except for few-sample scenarios, and thus is superior than previous subnetwork optimizations.

| Methods | SST-2 | QNLI |
|---|---|---|
| vanilla fine-tuning | 93.23 (93.69) | 92.31 (92.54) |
| Mixout | 93.45 (93.80) | 92.21 (92.45) |
| CHILD-TUNING$_D$ | 89.28 (94.04) | 92.36 (92.58) |
| DPS Dense | **93.77 (94.49)** | **92.61 (92.94)** |
| DPS Mix | 93.71 (94.16) | 92.45 (92.72) |

Table 5: Comparison between several fine-tuning methods and DPS with a sufficient number of training examples datasets. We report mean (max) development scores of 10 random seeds, and the best results are bolded.

### 4.4 Time Usage

| Methods | Vanilla | Mixout | R3F | R-Dropout | CHILD-TUNING$_D$ | DPS Dense | DPS Mix |
|---|---|---|---|---|---|---|---|
| Time Usage | x1.00 | x1.18 | x1.64 | x1.64 | x3.13 | x1.12 | x1.30 |

Table 6: Time usage for DPS and multiple fine-tuning regularization methods.

In this subsection, we investigate the computational efficiency of DPS. Specifically, we count the time usage of DPS in fine-tuning the $\text{BERT}_{LARGE}$ on RTE and compare time usage with various fine-tuning regularization methods, following [Lee et al., 2020]. We can notice from Table 6 that although DPS introduces a slight additional computational costs than vanilla fine-tuning, DPS is about 25% much faster than R3F and R-Dropout, and about **240%** much faster than CHILD-TUNING$_D$. We also discuss memory consumption for above methods in Appendix F.

## 5 Conclusion

In this work, we propose a family of new fine-tuning techniques to fine-tune large-scale PLMs adaptively: DPS Dense and DPS Mix, which respectively optimize full network update and stochastic parameter update processes. Our methods outperform various fine-tuning approaches in terms of stability and overall performance while saving computational costs a lot, and consistently improve the generalization ability of fine-tuned models by a large margin. Extensive results and analysis show that DPS can enhance largely on five different pre-trained language models meanwhile possessing extremely stable performance across a wider range of subnetwork sizes. We believe that training neural networks adaptively may have broader application scenarios. We leave this area of exploration for future work.

## 6 Acknowledgements

We thank all reviewers for their constructive comments. This research is supported by the National Key R&D Program under Grant No.2021ZD0110303, the National Natural Science Foundation of China under Grant No. 62072007, 62192733, 61832009, 62192731, 62192730.

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
