# Appendix for "Fine-Tuning Pre-Trained Language Models Effectively by Optimizing Subnetworks Adaptively"

**Haojie Zhang[1], Ge Li[1]∗, Jia Li[1], Zhongjin Zhang[1], Yuqi Zhu[1], Zhi Jin[1]**
[1]Key Laboratory of High Confidence Software Technologies (Peking University),
Ministry of Education; Institute of Software, EECS, Peking University, Beijing, China
`zhanghaojie@stu.pku.edu.cn, lige@pku.edu.cn`
`lijia@stu.pku.edu.cn, zjz123@stu.pku.edu.cn, zhuyuqi97@gmail.com, zhijin@pku.edu.cn`

## A    Appendix A. Case Study

In Sec.3.3, we have experimentally verified that DPS outperforms various fine-tuning methods. To understand what type of cases DPS predicts more accurately and justify the effectiveness of DPS from another perspective, we conduct case study. Specifically, we fine-tune BERT$_{LARGE}$ on RTE with 10 random restarts and count the overall proportion of easy cases (cases with more than 5 accurate predictions out of 10) and hard cases (cases with more than 5 predictions incorrect out of 10). Figure 1 summarizes the statistics. Compared with various baselines, DPS has the largest proportion of easy cases and the smallest proportion of hard cases. This demonstrates that compared with vanilla fine-tuning, Mixout, and CHILD-TUNING, DPS can better maintain general contextual representation, explore the potential value of data and models, and thus solve difficult cases without affecting the ability to identify easy cases, which is ultimately reflected in the improvement of metrics.

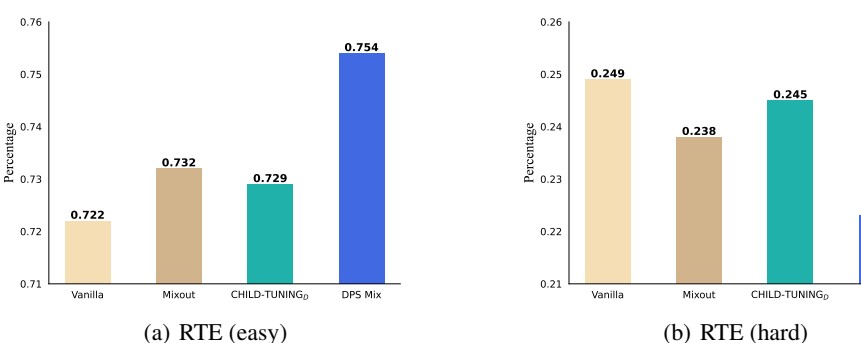

(a) RTE (easy)                                  (b) RTE (hard)

Figure 1: Subfigure.(a) summarizes the percentage of easy cases on various methods; Subfigure.(b) summarizes the percentage of hard cases on various methods.

## B    Appendix B. GLUE Benchmark Datasets

we conduct experiments on 8 datasets in GLUE benchmark Wang et al. [2019], the dataset statistics for each task are illustrated in Table 1. we also provide a brief description for each dataset:

- **RTE**: Binary entailment classification task Bentivogli et al. [2009]
- **MRPC**: Semantic similarity Dolan and Brockett [2005]

36th Conference on Neural Information Processing Systems (NeurIPS 2022).

- **STS-B**: Semantic textual similarity Cer et al. [2017]

- **CoLA**: Acceptability classification Warstadt et al. [2019]

- **SST-2**: Binary sentiment classification Socher et al. [2013]

- **QNLI**: Binary question inference classification Rajpurkar et al. [2016]

- **QQP**: Binary semantically equivalent classification Shankar Iyer and Csernai. [2017]

- **MNLI**: Textual entailment classification Williams et al. [2018]

| Dataset | RTE | MRPC | STS-B | CoLA | SST-2 | QNLI | QQP | MNLI |
|---|---|---|---|---|---|---|---|---|
| **Train Examples** | 2.5k | 3.7k | 5.7k | 8.5k | 67k | 105k | 364k | 393k |
| **Dev Examples** | 277 | 408 | 1.5k | 1.0k | 872 | 5.5k | 40k | 4.8k |
| **Metrics** | Acc | F1 | SCC | MCC | Acc | Acc | Acc | Acc |

Table 1: Eight datasets used in this paper form GLUE benchmark. Acc stands for Accuracy, SCC stands for Spearman Correlation Coefficient and MCC stands for Matthews Correlation Coefficient.

## C  Appendix C. Hyper-parameters and Experimental Details of Different Pre-trained Language Models

In this paper, we investigate the performance of DPS on five distinctive and widely used large-scale pre-trained language models, namely BERT Devlin et al. [2018], RoBERTa Liu et al. [2019], ELECTRA Clark et al. [2020], BART Lewis et al. [2019], and DeBERTa He et al. [2021]. BERT is the first Transformer Vaswani et al. [2017] encoder based pre-trained language model with Mask Language Modeling and Next Sentence Prediction pre-training tasks. RoBERTa is similar to BERT in terms of model architecture but is only pre-trained on the Mask Language Modeling task only, but for longer and on more data. ELECTRA is a BERT-like model trained to distinguish tokens generated by masked language model from tokens drawn from the natural distribution. BART is a sequence-to-sequence model trained as a denoising autoencoder. DeBERTa improves Transforme-based pre-trained model with disentangled attention mechanism and enhanced mask decoder. Table 2 summarizes the hyper-parameters of each model for each dataset. We use AdamWLoshchilov and Hutter [2019] optimizer, clip the gradients with a maximum norm of 1, and the maximum sequence length is set as 128. We use mixed precision training to speed up the experimental process. We conduct all the experiments on a single Tesla-V100 GPU (32G).

| Model | Datasets | Batch Size | Learning Rate | Training Epochs/Steps | Warmup Ratio/Steps | LLRD |
|---|---|---|---|---|---|---|
| BERT | all | 16 | 2e-5 | 3 epochs | 10% | - |
| RoBERTa | RTE | 16 | 2e-5 | 2036 steps | 122 steps | - |
| | MRPC | 16 | 1e-5 | 2296 steps | 137 steps | - |
| | STS-B | 16 | 2e-5 | 3598 steps | 214 steps | - |
| | CoLA | 16 | 1e-5 | 5336 steps | 320 steps | - |
| ELECTRA | RTE | 32 | 5e-5 | 10 epochs | 10% | 0.9 |
| | MRPC | 32 | 5e-5 | 3 epochs | 10% | 0.9 |
| | STS-B | 32 | 5e-5 | 10 epochs | 10% | 0.9 |
| | CoLA | 32 | 5e-5 | 3 epochs | 10% | 0.9 |
| BART | RTE | 32 | 1e-5 | 3 epochs | 10% | - |
| | MRPC | 64 | 2e-5 | 3 epochs | 10% | - |
| | STS-B | 32 | 2e-5 | 3 epochs | 10% | - |
| | CoLA | 64 | 2e-5 | 3 epochs | 10% | - |
| BEBERTA | RTE | 32 | 1e-5 | 6 epochs | 50 steps | - |
| | MRPC | 32 | 1e-5 | 6 epochs | 50 steps | - |
| | STS-B | 32 | 7e-6 | 4 epochs | 100 steps | - |
| | CoLA | 32 | 7e-6 | 6 epochs | 100 steps | - |

Table 2: Fine-tuning hyper-parameters of BERT and its variants as reported in the official repository of each model for best practice. Note that Layer-wise Learning Rate Decay (LLRD)Howard and Ruder [2018] is a method that applies higher learning rates for top layers and lower learning rates for bottom layers. This method is applied by ELECTRA when fine-tuning downstream tasks.

# D    Appendix D. Experimental Details for Different Fine-tuning Methods

The following is our hyperparameter search space for different fine-tuning regularization methods:

•**Mixout** We grid search Mixout probability $p \in \{0.1, 0.2, 0.3, 0.4, 0.5, 0.6, 0.7, 0.8\}$.

•**R3F**: We grid search Noise Types $\in \{\mathcal{N}, \mathcal{U}\}$, $\sigma \in \{1\text{e-}5\}$, $\lambda \in \{0.1, 0.5, 1.0, 5.0\}$.

•**Re-init**: We grid search $L \in \{1, 2, 3, 4, 5, 6, 7\}$.

•**CHILD-TUNING$_D$**: We grid search CHILD-TUNING$_D$ $p_D \in \{0.1, 0.2, 0.3\}$, and learning rate lr $\in \{2\text{e-}5, 4\text{e-}5, 6\text{e-}5, 8\text{e-}5, 1\text{e-}4\}$.

•**R-Dropout**: We grid search Dropout probability $p \in \{0.1\}$ and $\alpha \in \{0.1, 0.5, 1, 3, 5\}$.

•**DPS Dense**: We grid search reserved probability $p \in \{0.1, 0.2, 0.3, 0.4, 0.5\}$ and update ratio ur $\in \{0.05, 0.1, 0.2\}$.

•**DPS Mix**: We grid search reserved probability $p \in \{0.1, 0.2, 0.3, 0.4, 0.5\}$, and update ratio ur $\in \{0.05, 0.1, 0.2\}$.

# E    Appendix E. Mapping Strategy

We train DPS on SNLI and MNLI datasets respectively and evaluate several different target NLI datasets. The SNLI and MNLI datasets contain three labels: entailment, neutral, and contradiction, however, some datasets have only two labels (SciTial, RTE). The SciTail dataset contains two labels: entailment, neutral, and we map the predicted labels neutral and contradiction to neutral, following Mahabadi et al. [2021]. We conduct the same process for RTE.

# F    Appendix F. Memory Consumption

In addition to time usage, we further analyze memory consumption of various approaches when fine-tuning BERT$_{LARGE}$. The table below shows the results. DPS Dense requires extra memory because it needs to store Gradient Accumulation Matrix ($GAM$), DPS Mix requires more memory than DPS Dense because it needs to store Frequency Accumulation Matrix ($FAM$) and $GAM$. It is worth noting that since the size of $GAM$ and $FAM$ is fixed, the additional memory consumption introduced by DPS does not increase as the batch size increases. Therefore, as the batch size increases, the percentage of additional memory consumption decreases for DPS compared to vanilla fine-tuning. Overall, DPS does introduce additional memory overhead, but we believe that the memory consumption for DPS is acceptable compared to other regularization methods.

| Batch Size | Vanilla | Mixout | R3F | R-Dropout | CHILD-TUNING$_D$ | DPS Dense | DPS Mix |
|---|---|---|---|---|---|---|---|
| 16 | 10.6G | 12.9G | 17.7G | 17.7G | 10.9G | 16.3G | 21.1G |
| 32 | 14.3G | 16.3G | 28.3G | 28.3G | 14.7G | 20.0G | 24.7G |

Table 3:  Memory consumption for DPS and multiple fine-tuning regularization methods.