# OpenReview forum: "Fine-Tuning Pre-Trained Language Models Effectively by Optimizing Subnetworks Adaptively"
_NeurIPS.cc/2022/Conference — NeurIPS 2022 Accept_

### Official Review · Reviewer_BfEh · 2022-07-05

**Rating:** 7
**Confidence:** 4
**Soundness:** 3 good
**Presentation:** 3 good
**Contribution:** 2 fair

**Summary:**

Large pre-trained LMs form the baselines of many impressive results, but fine-tuning can, among other challenges, lead to overfitting and degradation of internal representations. This paper introduces Dynamic Parameter Selection, which selects subnetworks to perform staging updates based on gradients. The GLUE benchmark is used for evaluation, and results are positive.

**Questions:**

- why are so many (8) references needed at L17 to support the idea that pre-trained LMs are useful in NLP?
- while the limitations on pre-trained LMs in Sec 1 around overfitting and catastrophic forgetting are well known, how are the specific arguments (esp L34) related to those limitations?
- much of the information stored in neural networks concerns the relative differences between weights in the network — does altering only a subpart of a network affect not only its internal structure but its relationship with other information elsewhere in the network?
- As this approach is fairly generic, would it be straightforward to add vision datasets to those used in Sec 3.1?

**Limitations:**

Sec 4.4 is said to discuss limitations, though this is done only superficially, essentially only serving to further hype DPS.

**Strengths And Weaknesses:**

_Strengths_

- the two-stage process of accumulating in-batch gradients during fine-tuning, and subsequently using those gradients to derive stage-specific subnetworks to update is conceptually simple, and therefore clear.
- GLUE and the NLI datasets are appropriate, generic datasets.
- The models are very competitive, though in many cases the best means are within the indicated std (Table 1) therefore some statistical significance testing (correcting for multiple comparisons) is advised, as little can be conclusively concluded otherwise. The multitude of experimental scenarios (e.g., OoD generalization, low-resource scenarios) is a positive.


_Weaknesses_

- some minor spelling and grammatical mistakes (e.g., ‘has been prove’, L28; ‘to randomly replaces’, L30; ‘same sized’, L85; CHID-TUNING, L87, L92; ‘will not be update’, fig 1 caption) should be addressed. Along these lines, Figures 1 and 2 need to have their text made larger, for readability. References should be checked for proper formatting
- the treatment of related work (Sec 5) is quite superficial — more synthesis of the various approaches should be attempted.

---

> ### Author Response · Authors · 2022-08-01
> **Response to Reviewer BfEh**
>
> Thanks for your careful and valuable comments. We will answer your questions point by point.
>
> Q1: Some minor spelling and grammatical mistakes
>
> A1: Thanks for pointing out spelling and grammatical mistakes. We will correct them in the new version.
>
> Q2: The treatment of related work (Sec 5) is quite superficial — more synthesis of the various approaches should be attempted.
>
> A2: We agree with you that more synthesis of the various approaches should be attempted in Sec 5. We will adjust Sec 5 to synthesize more various approaches in the new version.
>
> Q3: Why are so many (8) references needed at L17 to support the idea that pre-trained LMs are useful in NLP?
>
> A3: Thanks for the suggestion, we will streamline the references at L17 in the new version.
>
> Q4: while the limitations on pre-trained LMs in Sec 1 around overfitting and catastrophic forgetting are well known, how are the specific arguments (esp L34) related to those limitations?
>
> A4: (1) Mixout and CHILD-TUNING are classic and competitive algorithms related to sub-network optimizations, which can effectively mitigate overfitting and catastrophic forgetting on pre-trained LMs. Since our paper is also exploring how to better fine-tune the pre-trained LMs utilizing sub-networks, our argument focuses on discovering the limitations of the existing sub-network optimizations, to explore more effective sub-network optimization algorithms. (2) Mixout updates sub-networks randomly, which is agnostic to the parameter importance; CHILD-TUNING updates fixed important sub-network, thus ignoring the changes in parameter importance that arise with the update process. Therefore, we propose DPS, an adaptive optimization algorithm that selects promising sub-networks evolving with the fine-tuning process.
>
> Q5: much of the information stored in neural networks concerns the relative differences between weights in the network — does altering only a subpart of a network affect not only its internal structure but its relationship with other information elsewhere in the network?
>
> A5: You have raised an interesting question. Yes, we believe that altering the sub-network also affects its relationship with the rest of the neural network. For DPS, we believe altering the more promising sub-network at the current stage will affect the relative importance of the neural network, which allows the model to clearly perceive the important sub-networks in subsequent training, thus making the model task-relevant. Overall, we believe that optimizing the stage-specific promising sub-networks adaptively will bring out those parts that are really important and stable, thus further improving the overall performance and stability.
>
> Q6: As this approach is fairly generic, would it be straightforward to add vision datasets to those used in Sec 3.1?
>
> A6: We believe this is a good question for future research. Considering that we have experimentally verified DPS's ability to effectively capture task-relevant parameters, we believe that DPS has a great potential to be useful on the vision datasets as well.

---

### Official Review · Reviewer_kBeJ · 2022-07-11

**Rating:** 7
**Confidence:** 3
**Soundness:** 3 good
**Presentation:** 3 good
**Contribution:** 3 good

**Summary:**

This paper proposes a Dynamic Parameter Selection (DPS) algorithm to address the overfitting and catastrophic forgetting issues in fine-tuning pre-trained language models.
The proposed DPS consists of a cyclic two-stage updating strategy:
1) accumulates the in-batch gradients of updated params during fine-tuning.
2) using the accumulated values in stage 1 to derive a subnetwork and keep non-subnetwork params constant.

The experiments are mainly carried out on the classification tasks of GLUE.
The proposed DPS is tested under various settings, and the results show that DPS shows consistent improvements over various baseline systems, which demonstrates the effectiveness of DPS.

**Questions:**

### Major comments
1. The experiments are only carried out on classification tasks. Did you apply your method to other tasks, such as machine reading comprehension (MRC), or named entity recognition (NER)?
2. In Section 4.4, the authors have demonstrated that the proposed DPS is more efficient than other methods in time usage. How about memory consumption?
3. line 187-188: Did you arbitrarily choose the size (8K) of subsampled MNLI/SNLI?
4. It would be nice if the authors could explicitly discuss and compare two variants of DPS along with the experimental results.
5. Section 3.5: If I understand correctly, all experiments are carried out on the same subsets (either 0.5k or 1k) of the tasks in GLUE. It would be nice if the authors could release what the exact subsets are sampled to enable the researchers to compare the results in the future.


### Minor comments
1. line 63: I know what is "std," but please use its full name when it first appears in a formal paper.
2. line 226: DEBERTA -> DeBERTa
3. Figure 1: it seems that "us" is a single variable. Thus, "2kus+i" seems to be quite confusing (might be interpreted as 2*k*u*s). Please try to make the naming of the variable more clear.

**Limitations:**

The authors claim in the checklist that they have discussed their limitation in Section 4.4, noting the additional training cost of their method. However, the Section 4.4 seems not to be discussing their limitation but advantages over other methods. The authors are encouraged to discuss some "true" limitations of their method. For example, does your model consume more memory than other methods? Please try to take a balanced view in interpreting the limitation of your method.


**Strengths And Weaknesses:**

### Strengths
1. The proposed method addresses an important topic of the overfitting and catastrophic forgetting issues in fine-tuning pre-trained language models.
2. The formulation of the proposed DPS is technically sound and reasonable.
3. The experiments are carried out in various aspects to fully demonstrate the effectiveness of the proposed model.


### Weaknesses
1. The experiments are only carried out on classification tasks. Using one more type of NLU task would have further strengthened the results.
2. The discussion of the limitation is not informative.

---

> ### Author Response · Authors · 2022-08-01
> **Response to Reviewer kBeJ**
>
> Thanks for your careful and valuable comments. We will answer your questions point by point.
>
> Q1: Did you apply your method to other tasks, such as machine reading comprehension (MRC), or named entity recognition (NER)?
>
> A1: We consider classification tasks to be basic and representative tasks. Limited by space constraints, we only experimented on the classification tasks, following several previous studies focusing on fine-tuning instability and generalizability (e.g., Mixout, CHILD-TUNING, and Re-init).
>
> Q2: In Section 4.4, the authors have demonstrated that the proposed DPS is more efficient than other methods in time usage. How about memory consumption?
>
> A2: We agree with you that discussion on memory consumption for the proposed DPS is useful for the readers. In addition to time usage, we further analyze memory consumption of various approaches when fine-tuning BERT$_{LARGE}$. The table below shows the results. DPS Dense requires extra memory because it needs to store Gradient Accumulation Matrix (GAM), DPS Mix requires more memory than DPS Dense because it needs to store Frequency Accumulation Matrix (FAM) and GAM. It is worth noting that since the size of GAM and FAM is fixed, the additional memory consumption introduced by DPS does not increase as the batch size increases. Therefore, as the batch size increases, the percentage of additional memory consumption decreases for DPS compared to vanilla fine-tuning. Overall, DPS does introduce additional memory overhead, but we believe that the memory consumption for DPS is acceptable compared to other regularization methods.
> | batch size| Vanilla     |  Mixout    | R3F     | R-Dropout     | CHILD-TUNING$_{D}$     | DPS Dense    |DPS Mix     |
> | -------- | -------- | -------- | -------- | -------- | -------- | -------- |-------- |
> | 16| 10.6G | 12.9G | 17.7G | 17.7G | 10.9G  | 16.3G |21.1G|
> | 32| 14.3G | 16.3G |  28.3G | 28.3G | 14.7G  | 20.0G |24.7G|
>
> Q3: line 187-188: Did you arbitrarily choose the size (8K) of subsampled MNLI/SNLI?
>
> A3: Yes. We choose the 8K size randomly.
>
> Q4: It would be nice if the authors could explicitly discuss and compare two variants of DPS along with the experimental results.
>
> A4:  (1) Overall, both DPS Dense and DPS Mix can obtain decent improvement in few-sample, low-resource, and out-of-domain settings. Compared to DPS Dense, DPS Mix tends to achieve lower standard deviations and is therefore more stable. (2) When the training examples are sufficient, DPS Dense is a more competitive approach to improve the upper limit of the model compared to DPS Mix.
>
> Q5: Section 3.5: If I understand correctly, all experiments are carried out on the same subsets (either 0.5k or 1k) of the tasks in GLUE. It would be nice if the authors could release what the exact subsets are sampled to enable the researchers to compare the results in the future.
>
> A5: (1) Yes, all experiments are carried out on the same subsets (either 0.5k or 1k) of the tasks in GLUE. (2) Thanks for the suggestion, we will release the exact subsets to our repository for better reproduction and comparison.
>
> Q6: Minor comments
>
> A6: Thanks for pointing out {1.,2.,3.} in minor comments, we will correct all of them in the new version.

---

### Official Review · Reviewer_sChX · 2022-07-12

**Rating:** 7
**Confidence:** 4
**Soundness:** 3 good
**Presentation:** 4 excellent
**Contribution:** 3 good

**Summary:**

This paper proposed a new large-scale language model fine-tuning algorithm called Dynamic Parameter Selection (DPS). This algorithm essentially uses a two stage cyclic updating strategy, where in Stage-1 it accumulates in-batch gradients of updated parameters during fine-tuning, and in Stage-2, it derives a subnetwork based on the Stage-1 accumulated gradients and their relative importance. The paper has proposed two variants of the algorithm called DPS Dense and DPS Mix, and show that it consistently performs better than previous approaches on sub-network finetuning strategies. Also, the experimental results on the out-of-domain show the usefulness of DPS.


**Questions:**

1) In Table-1, what is the reason to choose only four tasks from GLUE and why only these tasks?
2) Suggestion: Maybe it is better to merge Section-5 with Section-2.
3) In all the ablations, comparison is based on vanilla fine-tuning, can you also add any sub-network optimization model as well?


**Strengths And Weaknesses:**

Strengths:
1) The paper is well written and easy to follow.
2) The proposed DPS model is interesting and novel with better results than other algorithms related to sub-network optimization.
3) The results on out of domain generalization are very exciting. However, it is not clear why this approach leads to better generalization.
4)  The ablations are very thorough and useful such as experiments on low resource scenario, effect of subnetwork size, different pre-trained language models, and DPS with sufficient number of training examples.

Weaknesses:
1) It would have been more interesting if the paper could show any theoretical guarantees or understanding of why this algorithm is superior over other sub-network optimizations.
2) It would be interesting to check whether the observations hold for more than the current 4 tasks reported in Table-1. Is there any reason for not showing on a complete GLUE dataset?

---

> ### Author Response · Authors · 2022-08-01
> **Response to Reviewer sChX**
>
> Thanks for your careful and valuable comments. We will answer your questions point by point.
>
> Q1: In Table-1, what is the reason to choose only four tasks from GLUE and why only these tasks?
>
> A1: (1) Following recent competitive few-sample fine-tuning approaches: Mixout, CHILD-TUNING, and Re-init, we compare proposed approaches with various fine-tuning regularization methods on RTE, MRPC, STS-B, and CoLA from GLUE. (2) One of the purposes of this paper is to improve the stability and overall performance on datasets with few samples. Since Phang et al. (2018) observed fine-tuning instability focusing on datasets with samples less than 10K, we choose datasets with training samples less than 10K from GLUE (RTE, MRPC, STS-B, and CoLA) to validate the effectiveness of DPS.
>
> Q2: Suggestion: Maybe it is better to merge Section-5 with Section-2.
>
> A2: Thanks for the suggestion, we will adjust Section-5 in the new version.
>
> Q3: In all the ablations, comparison is based on vanilla fine-tuning, can you also add any sub-network optimization model as well?
>
> A3: (1) In Section 4.1, we experimentally verify that DPS can capture more task-relevant parameters at different sub-network sizes compared to other sub-network optimizations. (2) In Section 4.2, we only compare with vanilla fine-tuning because (a) we have shown in Table 1 that DPS outperforms various competitive methods in terms of overall performance and stability.  In Table 4, we only demonstrate that DPS can adapt well to different pre-trained language models; (b) following CHILD-TUNING and Re-init, we compare various methods on BERT$_{LARGE}$, and verify the effectiveness of the proposed methods compared to vanilla fine-tuning at different pre-trained models. (3) In Section 4.3, we further compare DPS with Mixout and CHILD-TUNING in the table below.
> | Methods| SST-2|QNLI|
> | -------- | -------- | -------- |
> | vanilla fine-tuning| 93.23 (93.69)  | 92.31 (92.54)|
> | Mixout| 93.45 (93.80)  |92.21 (92.45)|
> | CHILD-TUNING$_{D}$|89.28 (94.04) |92.36 (92.58)|
> | DPS Dense|93.77 (94.49)|92.61 (92.94)|
> | DPS Mix|93.71 (94.16) |92.45 (92.72)|
>
> [1] Jason Phang, Thibault F´evry, and Samuel R Bowman. Sentence encoders on stilts: Supplementary training on intermediate labeled-data tasks. arXiv preprint arXiv:1811.01088, 2018.

---

### Official Review · Reviewer_SWtE · 2022-07-18

**Rating:** 6
**Confidence:** 4
**Soundness:** 2 fair
**Presentation:** 2 fair
**Contribution:** 2 fair

**Summary:**

The motivation behind this paper leaves from current limitations of fine-tuning, namely fine-tuning instability and poor generalizability (particularly in OOD settings). To address these limitations, the authors propose a Dynamic Parameter Selection (DPS) algorithm, which fine-tunes pre-trained networks adaptively in two stages: 1) Stage I accumulates gradients of updated parameters during fine-tuning, and 2) Stage II derives a subnetwork composed of the important parameters to update based on the accumulated values in Stage I. The authors propose two versions of their algorithm, DPS Dense (which optimizes the entire network update process) and DPS Mix (which optimizes the stochastic parameter update process).

The authors conduct experiments in OOD setting and low-resource scenarios on GLUE and NLI datasets, showing their proposed method outperforms prior approaches. On the GLUE benchmark, DPS gains range between 0.44-1.33 points and 0.29-0.8- std. Moreover, the authors conduct ablation experiments where they analyze performance of DPS when varying the subnetwork size, using various pre-trained models, sufficient number of training examples and time duration. The authors conclude that their proposed DPS method leads to stable improvement in pre-trained model performance with sufficient number of training examples, is time-efficient and captures task-relevant parameters across a variety of subnetwork sizes.


**Questions:**

Line 175: “DPS Dense outperforms all baselines” - from the results in Table 1, DPS Dense does not seem to outperform all baselines, particularly on MRPC and STS-B datasets. While on average it does, I would recommend clarifying this aspect in the presentation of the results.

Lines 177-178: “DPS saves 177 additional computational overhead imposed by computing Fisher Information based on all training 178 samples”

Regarding the results in Table 2 (Out-of-Domain Generalization), how do you explain that DPS Dense is performing better than DPS mix on the MNLI dataset, while results on the SNLI dataset seem to show the opposite trend that DPS Dense outperforms DPS Mix? What about the standard deviation?

Table 4: Have any of the pre-trained models been trained on the datasets you are evaluating their performance on?

Section 4.3: “two datasets in GLUE with training samples > 50K” - which is the motivation behind selecting datasets with > 50K in particular? The paper does not provide an explanation for your particular choice of > 50K.


**Limitations:**

The work seems largely incremental in nature, and the methods proposed by the authors are largely inspired by previous work in the literature. The authors nevertheless convincingly show the good performance of their method.

The paper lacks a comprehensive discussion of the limitations of the proposed methods, the computational overhead introduced. While Section 4.4 talks about the slight computational costs compared to vanilla fine-tuning, this is only in time usage and not in computational complexity.

Line 80: typo - “Differ from fine-tuning whole network” (--> different)



The paper addresses a relevant problem on how to more efficiently fine-tune pretrained models, particularly in data-limited regime and out of distribution settings. The method proposed by the authors is inspired by previous work, nevertheless empirical results demonstrate it performs well. The paper would benefit from a more clear discussion of the limitations of the proposed approach, particularly in terms of the computational complexity and the need to first do fine-tuning before applying the proposed approached (DPS Dense), as well as the additional data structures needed to store the frequency of each parameter being update (DPS Mix).


**Strengths And Weaknesses:**

Strengths:
- The intuition behind the idea that fine-tuning a subnetwork instead of the entire network can yield more relevant parameters for a specific downstream task is valuable and does make a lot of sense


- The authors use an extensive set of baselines from the literature to compare their proposed method with, including Mixout (Lee et al, 2020), R3F (Aghanjanyan et al, 2021), R-Dropout (Wu et al, 2021), Child-tuning (Xu et al, 2021), and Re-init (Zhang et al, 2021)


- The authors convincingly demonstrate that their method performs well in Out-of-distribution and low-resource scenarios, and the ablation experiments show that the proposed method works when combined with different pre-trained models

Weaknesses:
- DPS dense needs a first pass in Stage I that finetunes the whole network for gradient accumulation. While I understand the need for this pass, it seems wasteful, why would someone use DPS on top of fine-tuning the network? Unfortunately the authors do not discuss this limitation of their proposed method in the paper.


- While DPS Mix is proposed as an alternative to optimize the fine-tuning process, it requires a Frequency Accumulation Matrix (FAM) to store the frequency of every parameter being updated, in addition to the Gradient Accumulation Matrix (GAM) required by DPS Dense. The authors do not discuss the computational bottleneck and complexity of these design choices. Moreover, Stage I of DPS Mix is identical to previously proposed methods in the literature (Mixout).


- The authors are mainly comparing the obtained results with ChildTuning, without motivating why they focus particularly on this baseline method.

---

> ### Author Response · Authors · 2022-08-01
> **Response 1 to Reviewer SWtE**
>
> Thanks for your careful and valuable comments. We will answer your questions point by point.
>
> Q1: DPS dense needs a first pass in Stage I that finetunes the whole network for gradient accumulation. While I understand the need for this pass, it seems wasteful, why would someone use DPS on top of fine-tuning the network?
>
> A1: (1) Stage I is needed because (a) it can effectively combine two processes: fine-tuning and deriving sub-networks, which greatly reduces the additional heavy computational costs introduced by separating above two processes like ChildTuning; (b) it provides a reasonable basis for Stage II to update more promising sub-networks based on the changing parameter importance during fine-tuning, while adjusting the parameter distributions. (2) Although DPS requires two stages, it is a more time-efficient approach than several fine-tuning methods (as analyzed in section 4.4). We also experimentally verify that DPS outperforms multiple fine-tuning regularization methods in terms of overall performance and stability. So we believe that DPS is a more competitive alternative.
>
> Q2: Discussion on computational bottleneck and complexity
>
> A2: We agree with you that more discussion on the computational bottleneck is necessary. In addition to time usage, we further analyze memory consumption of various approaches when fine-tuning BERT$_{LARGE}$. The table below shows the results. DPS Dense requires extra memory because it needs to store Gradient Accumulation Matrix (GAM), DPS Mix requires more memory than DPS Dense because it needs to store Frequency Accumulation Matrix (FAM) and GAM. It is worth noting that since the size of GAM and FAM is fixed, the additional memory consumption introduced by DPS does not increase as the batch size increases. Therefore, as the batch size increases, the percentage of additional memory consumption decreases for DPS compared to vanilla fine-tuning. Overall, DPS does introduce additional memory overhead, but we believe that the memory consumption for DPS is acceptable compared to other regularization methods.
> | batch size| Vanilla     |  Mixout    | R3F     | R-Dropout     | CHILD-TUNING$_{D}$     | DPS Dense    |DPS Mix     |
> | -------- | -------- | -------- | -------- | -------- | -------- | -------- |-------- |
> | 16| 10.6G | 12.9G | 17.7G | 17.7G | 10.9G  | 16.3G |21.1G|
> | 32| 14.3G | 16.3G |  28.3G | 28.3G | 14.7G  | 20.0G |24.7G|
>
> Q3: The authors are mainly comparing the obtained results with ChildTuning, without motivating why they focus particularly on this baseline method.
>
> A3: (1) ChildTuning and Mixout are classic and competitive algorithms related to sub-network optimizations, while the contribution of our DPS is to better adapt the model utilizing sub-networks. Thus, we make various comparisons with ChildTuning and Mixout. (2) Among all baselines mentioned in section 3.3.1, only ChildTuning conducts experiments in OOD setting. Considering we are also concerned about this challenge, we compare DPS with ChildTuning in OOD setting. (3) We compared DPS with other competitive fine-tuning methods in section 3.3.2.
>
> Q4: Line 175: “DPS Dense outperforms all baselines”.
>
> A4: Thanks for pointing out this inaccurate description, we will carefully correct them in the new version.
>
> Q5: Lines 177-178: “DPS saves 177 additional computational overhead imposed by computing Fisher Information based on all training 178 samples”
>
> A5: ChildTuning utilizes Fisher Information to derive important sub-network based on all training samples, which is conducted outside the fine-tuning process and requires heavy extra time overhead. DPS accumulates in-batch gradients to derive important sub-networks during fine-tuning, and thus the time introduced by DPS is limited. Therefore, we consider DPS to be more time-efficient. We feel much sorry for any confusion and will present this more clearly in the new version.
>
> Q6: How do you explain that DPS Mix is performing better than DPS Dense on the MNLI dataset, while results on the SNLI dataset seem to show the opposite trend that DPS Dense outperforms DPS Mix? What about the standard deviation?
>
> A6: (1) In MNLI, ChildTuning and DPS Mix outperform vanilla fine-tuning and DPS Dense respectively. In SNLI, ChildTuning only has negligible improvement than vanilla fine-tuning, and the improvement of DPS Mix is not as obvious as it of DPS Dense. Considering that ChildTuning and DPS Mix utilize pre-trained weights, we think that it is because on MNLI, the model relies more on pre-trained weights than on SNLI. (2) Although DPS Dense and DPS Mix perform differently on MNLI and SNLI in OOD setting, they consistently outperform vanilla fine-tuning and ChildTuning on both MNLI and SNLI overall. (3) In OOD setting, the average standard deviations of the different approaches are comparable.

---

> ### Author Response · Authors · 2022-08-01
> **Response 2 to Reviewer SWtE**
>
> Q7:Have any of the pre-trained models been trained on the datasets you are evaluating their performance on?
>
> A7: To ensure a fair comparison, all experiments in this paper are conducted directly on the open-source pre-trained models provided by HuggingFace.
>
> Q8: Which is the motivation behind selecting datasets with > 50K in particular?
>
> A8: Phang et al. (2018) observed fine-tuning instability focusing on datasets with samples less than 10K. Selecting datasets more than 50K is to indicate that we are using much larger datasets than 10K to verify the performance of DPS with sufficient training samples.
>
> [1] Jason Phang, Thibault F´evry, and Samuel R Bowman. Sentence encoders on stilts: Supplementary training on intermediate labeled-data tasks. arXiv preprint arXiv:1811.01088, 2018.

---

### Meta-Review · Area_Chair_6iUo · 2022-08-24

**Recommendation:** Accept
**Confidence:** Certain

**Metareview:**

The reviewers were very positive about this paper, which proposes a method for selective fine-tuning of language models on downstream tasks. The method works well in various settings including realistic out-of-distribution and low-resource scenarios.

The main strengths are:
* useful idea
* conceptually simple method
* the choice of datasets
* the use of competitive models
* convincing experiments in multiple settings, including OOD generalization

The reviewers also noted some weaknesses, especially:
* the need for a first pass, computational bottleneck, --> author response seems satisfactory
* the choice of baseline to compare against
* need for theoretical guarantees  --> to me, this is not strictly necessary for an empirical paper
* experiments only with classification tasks --> to me, the choice of tasks (GLUE, NLI) is reasonable, as the authors argue
* uninformative discussion of limitations

The authors did a fine job addressing most of these issues in their response and they should update their paper accordingly. Concerning the last point: the authors did not address this point, and are strongly encouraged to do so in their revision.

**Award:**

No

---

### Decision · Program_Chairs · 2022-09-14

Accept